# *Hermetia illucens* Larvae Meal Enhances Immune Response by Improving Serum Immunoglobulin, Intestinal Barrier and Gut Microbiota of Sichuan White Geese After Avian Influenza Vaccination

**DOI:** 10.3390/vetsci11120615

**Published:** 2024-12-02

**Authors:** Yufei Xie, Yongfeng Hao, Fuxing Gui, Xifeng Li, Huan Huang, Pingrui Yang, Chonghua Zhong, Liting Cao

**Affiliations:** 1Department of Traditional Chinese Veterinary Medicine, College of Veterinary Medicine, Southwest University, Chongqing 402460, China; hezhangweiqi@163.com (Y.X.); 18382034193@163.com (F.G.); lixiaohei2515@163.com (X.L.); huanghuan128@126.com (H.H.); pingruiyang@outlook.com (P.Y.); 2College of Animal Science and Technology, Chongqing Three Gorges Vocational College, Chongqing 404155, China; h1148764878@163.com; 3Chongqing Rongchang Vocational Education Center, Chongqing 402460, China; diana_200899@163.com

**Keywords:** *Hermetia illucens* Larvae Meal, Sichuan White Geese, avian influenza vaccination, intestinal mucosal immunity, intestinal microflora

## Abstract

It has been observed that HILM enhances growth performance and immune function in poultry, yet the effects and mechanisms on geese remain unclear. Experiment I of this study was conducted to evaluate the optimal amount of HILM dietary addition by measuring growth performance and AIV antibody in 64 Sichuan White Geese vaccinated against avian influenza. Experiment II explored the underlying mechanisms by measuring serum immunoglobulins, immune-related gene expression, intestinal morphology, barrier-related gene expression and gut microbiota at the optimal dose. This study revealed the effects and mechanisms of HILM on geese and provided insights into using HILM in geese husbandry.

## 1. Introduction

Global population growth has led to shifts in food consumption habits, lifestyles and dietary preferences, driving up the demand for animal proteins [1]. Projections indicate that the consumption of animal products will increase by 60–70% by 2050, which will put pressure on resources such as land, water and energy for the production of feedstock for animal and plant feed proteins [2]. To address this challenge, novel, cost-effective alternatives or supplements need to be explored to facilitate the growth of livestock and poultry production [3]. Insects emerge as promising choices because they are nutrient-rich and consume less feeding material [4]. Studies have shown that insects have good nutritional value for humans, poultry, pigs and fish, the consumption of insects has nutritional, environmental and economic benefits, and insect products have been proven to benefit the health and welfare of livestock and poultry [5]. Additionally, the incorporation of insects into feed could potentially reduce the need for antibiotics in the breeding industry [6].

*Hermetia illucens* is an insect of the Diptera family whose larval meal is rich in proteins, fats and minerals, and its protein has a good balance of essential and non-essential amino acids [7]. Dietary proteins could regulate the composition and metabolic activity of the gut microbiota, thereby affecting the health of animals. Recently, *Hermetia illucens* larval meal (HILM) has been added to the diets of carnivorous aquatic animals, poultry, swine and companion animals [8]. Related studies have demonstrated that supplementation of 2% HILM to the diets of weaned piglets up-regulated the mRNA expression of developmental and anti-inflammatory cytokine genes, promoted the concentration of SIgA in the ileal mucosa and increased the abundance of probiotics such as *Lactobacillus* and *Bifidobacterium* in the cecum [9]. Another study also found that dietary supplementation with 4% HILM protected the integrity of ileal villi in the presence of enterotoxin-producing *Escherichia coli K88 (E. coli K88)* infection of piglets, improved the expression of intestinal mucosal tight junction proteins and the abundance of *Lactobacillus* and decreased the abundance of *Streptococcus*, thereby improving the intestinal health of weaned piglets [10]. Replacement of 15% soybean meal crude protein with HILM in broiler diets had no significant effect on performance and health; however, when substitution reached 30%, performance was impaired, and ileal crude protein and amino acid digestibility was decreased [11].

The main active components of HILM include antimicrobial peptides, lauric acid and chitin, among which the antimicrobial peptides have outstanding antibacterial efficacy, low risk of bacterial resistance and may have a potential anti-infective effect against fungi and viruses [12,13]. Moreover, the lauric acid in HILM biomass showed antimicrobial activity both in vivo and in vitro and has no toxicity to humans or animals, nor does it cause residues and cross-resistance induction [14]. Similarly, chitin and chitosan of *Hermetia illucens* exoskeletons stimulate the innate immune system, possess antimicrobial properties and inhibitory effects against harmful Gram-negative bacteria and positively affect the growth of beneficial microorganisms [15]. A novel polysaccharide from *Hermetia illucens* has also been found to activate innate immunity in mammalian macrophages [16]. To sum up, HILM can not only provide an alternative to traditional protein and energy sources in animal diets but also has the potential to supersede antibiotics and prebiotics in feed.

Avian influenza virus (AIV) has always posed a threat to animal and human health. Avian influenza in geese is mainly a respiratory infection that has caused large numbers of geese deaths and consequent severe economic losses [17]. Improving the immune response of AIV vaccination during farming is the key measure to prevent and control AIV. Therefore, the aim of this study was to investigate the immune-enhancing effects of HILM, whose findings will provide some insights into the application of HILM in geese or other waterfowl.

## 2. Materials and Methods

### 2.1. Animal Ethics

This animal study was conducted in accordance with the guidelines approved by the Southwest University Laboratory Animal Welfare Ethics Committee, Chongqing, China. (Ethics NO.: IACUC-20210905-01).

### 2.2. Preparation of Test Subjects

The Sichuan White Geese were purchased from Chongqing Qingshuiwan Breeding Goose Industry Co., Ltd. (Chongqing, China). HILM was purchased from Zhengzhou BENNONG-TEGH Co., Ltd. (Zhengzhou, China), and the results of its nutrient content are shown in Table 1. The basal diets were purchased from Chongqing Huiguang Feed Co., Ltd. (Chongqing, China), and formulated according to the requirements of the National Research Council (1994), and their nutritional values are shown in Table 2. During the experiment, all geese were housed in pens, fed and watered freely and vaccinated with trivalent inactivated Recombinant avian influenza virus (H5 + H7) trivalent inactivated vaccine (Harbin Pharmaceutical Group Biological Vaccine Co., Ltd. (Harbin, China), Production lot number: 202279) at 12 and 33 d of age, respectively.

### 2.3. Experimental Design

The experiment consisted of two parts, Experiment I and Experiment II.

Experiment I: A total of 64 1-day-old Sichuan White Geese were randomly divided into 4 experimental groups with 4 replicates per group and 4 goslings per replicate. The control group was fed the basal diet. At the same time, the experimental groups were fed the basal diet supplemented with 1%, 2% and 4% HILM for 40 days. During the experiment, all experimental geese were weighed every other week, and the daily feed intake was accurately recorded. Blood samples were collected aseptically at 7, 14 and 21 d after the 1st immunization and 7 d after the 2nd immunization. Serum samples were separated and stored in a −80 °C refrigerator for serum biochemistry, AIV antibody titers and specific antibody. At the end of the experiment, all the geese were sacrificed by isoflurane anesthesia, and the bursa of Fabricius, spleen and thymus were carefully separated, dried with absorbent paper and weighed to calculate the immune organ index.

Experiment II: A total of 32 1-day-old Sichuan White Geese were randomly divided into a control group and 1% HILM group with 4 replicates per group and 4 goslings per replicate. The control group was fed the basal diet, and the experimental group was fed the basal diet supplemented with 1% HILM for 40 days. After serum samples were collected at the end of the experiment, all the experimental geese were sacrificed by isoflurane anesthesia, and spleen, duodenum, jejunum, ileum and cecal contents of the same location and size were accurately collected. Among them, the tissues of the duodenum, jejunum and ileum were fixed in 4% paraformaldehyde for HE staining. Spleen tissue, remaining jejunal tissue and cecal contents were placed in 2 mL EP tubes, flash-frozen in liquid nitrogen, and stored in a −80 °C refrigerator for the detection of splenic immune factors, jejunal barrier gene expression levels and 16S sequencing.

### 2.4. Growth Performance

In experiment I, the geese were weighed on an empty stomach every week, and the total amount of feed per cage and the remaining amount of feed were recorded the next day. At the end of the experiment, average daily feed intake (ADFI), average daily gain (ADG), and feed conversion ratio (FCR) were calculated to evaluate the growth performance.

### 2.5. Immunological Potency and Specific Antibody of AIV

In experiment I, HI and HA tests were used to detect the antibody titers of all the geese at 7, 14 and 21 d after the 1st immunization and 7 d after the 2nd vaccination. At the same time, an ELISA kit was used to detect the AIV-specific antibody in serum.

### 2.6. Serum Biochemical Indicators and Jejunum SIgA

In experiment II, serum IgG, IgG1, IgG2a, IgG2b, IgG3, complement C3, complement C4 and jejunum SIgA were measured using ELISA kits according to the supplier’s instructions. The kits used were from the Nanjing Jiancheng Bioengineering Institute, Nanjing, China (Product lot Number: IgG, BPE60414; IgG1, BPE60376; IgG2a, BPE60382; IgG2b, BPE60383; IgG3, BPE60388; C3, H186-1-1; C4, H186-2-1; SIgA, BPE60153).

### 2.7. Intestinal Morphology Analysis

In experiment II, intestinal tissues from the same intestinal segment with a thickness of no more than 3 mm preserved in 4% paraformaldehyde were collected, thoroughly washed in water, subjected to a series of careful dehydration procedures and embedded in paraffin. The embedded tissues were cut into 4–7 μm thick slices and stained with hematoxylin-eosin, and the tissue structure was observed under a light microscope. IMAGE J was used to measure the length of intestinal villi (VH) and the depth of crypts (CD), and the ratio of villus length to crypt depth (VH/CD) was calculated.

### 2.8. mRNA Expression Spleen Immune Genes and Jejunum Barrier Gene

In experiment II, total RNA was extracted from the spleen and jejunum using the Trizol method, and the purity and concentration of total RNA were determined by Nanodrop 2000. The total RNA with an OD value between 1.8 and 2.0 was used for subsequent experiments. The cDNA was synthesized by reverse transcription according to the instructions of the PrimeScript™ RT Reagent Kit with gDNA Eraser (Perfect Real Time). β-Actin was used as the internal reference gene, and IL-1β, IL-2, IL-6, IL-10, TNF-α, IFN-γ, CD4, CD8a, TGF-β, ZO-1, Occludin and Claudin-1 were used as target genes. NCBI Primer Blast was used to design the primers for RT-qPCR, and the primer sequences are shown in Table 3. The primers were synthesized by Sangon Biotech Co., Ltd. (Shanghai, China). The RT-qPCR reaction system was prepared according to the instructions of the TB Green^®^ Premix Ex Taq™ II kit (Tli RNaseH Plus), and the RT-qPCR reaction system was 10 μL: TB Green Premix Ex Taq II (Tli RnaseH Plus) 5 μL, PCR forward primer 0.4 μL, PCR reverse primer 0.4 μL, template DNA 1 μL, Rox reference dye 0.2 μL, ddH2O 3 μL. Reaction conditions: predenaturation at 95 °C for 3 min, denaturation at 95 °C for 15 s, annealing at 56 °C for 30 s, extension at 72 °C for 30 s, 40 cycles. All samples were performed in triplicates. Relative expression was calculated by the 2^−∆∆Ct^ quantitative analysis method (PrimeScript™ RT Reagent Kit with gDNA Eraser, Code No., RR047A; TB Green^®^ Premix Ex Taq™ II kit, Code No., RR820A, all purchased from Takara Biomedical TechnologyCo., Ltd. (Beijing, China)).

### 2.9. Gut Microbiota Analysis

Total genomic DNA from the microbial community of each sample was extracted using the E.Z.N.A. Soil DNA Kit (Omega Bio-tek, Inc., Norcross, GA, USA) according to the manufacturer’s instructions. The concentration and quality of the genomic DNA were measured using a NanoDrop 2000 spectrophotometer (Thermo Fisher Scientific Inc., Waltham, MA, USA). Universal primer 338F (5′-ACTCCTACGGGAGGCAGCAG-3′) and 806R (5′-GGACTACNNGGGTATCTAAT-3′) were used to amplify the V3-4 hypervariable region of the bacterial 16S rRNA gene. These primers incorporated sample-specific barcode sequences (provided by Allwegene Company, Beijing, China). The PCR was performed on the ABI 9700 PCR instrument (Applied Biosystems, Providence, RI, USA), and the PCR products were purified using the Agencourt AMPure XP Kit (Beckman Coulter, Inc., Brea, CA, USA). Sequencing libraries were generated using an NEB Next Ultra II DNA Library Prep Kit (New England Biolabs, Inc., Ipswich, MA, USA) following the manufacturer’s recommendations. Deep sequencing was performed on the Illumina Miseq/Nextseq 2000/Novaseq 6000 (Illumina, Inc., San Diego, CA, USA) platform at Allwegene Technology Co., Ltd. (Beijing, China). After the run, image analysis, base calling and error estimation were performed using Illumina Analysis Pipeline Version 2.6 (Illumina, Inc., USA).

Qualified sequences were clustered into operational taxonomic units (OTU) at a similarity threshold of 97% using the Uparse algorithm of Vsearch (v2.7.1) software. QIIME (v1.8.0) was used to generate rarefaction curves, calculate the richness and diversity indices based on the OTU information and use R (v3.6.0) software to plot. The β-Diversity distance matrix between samples was calculated using the Bray–Curtis algorithms and plotted PCoA with the Arithmetic Mean clustering tree. Moreover, the relative abundance of species at different taxonomies was also obtained, using Python (v2.7) software for LEfSe analysis.

### 2.10. Data Analysis

Data were presented as the Mean ± SEM and analyzed statistically using GraphPad Prism 8.0 (GraphPad Software Inc., Chicago, IL, USA). One-way ANOVA and T-test were used to determine the significant difference among groups. A value of *p* > 0.05 was considered no significant difference, *p* < 0.05 was statistically significant, and *p* < 0.01 was considered an extremely significant difference.

## 3. Results

### 3.1. Effects of HILM on Growth Performance

As shown in Table 4, 1% HILM supplementation significantly increased ADG (*p* < 0.05) compared with the control group, while 2% and 4% HILM also increased ADG, but the difference was not statistically significant (*p* > 0.05).

### 3.2. Effects of HILM on AIV Immunological Potency and Specific Antibody

As shown in Figure 1, compared with the control group, the addition of 1% HILM significantly increased the serum antibody titers of H5-R14 (Figure 1b) and H7-R4 (Figure 1c) at 33 d (*p* < 0.01 or *p* < 0.05) and extremely significantly increased the serum antibody titers of H5-R13 (Figure 1a) at 40 d (*p* < 0.01). In addition, H5-R14 (Figure 1b) serum antibody titers were also significantly increased at 33 d in the 2% HILM group (*p* < 0.05).

As shown in Figure 2, compared with the control group, the AIV-specific antibodies in the serum of 1% HILM showed an upward trend compared with the control group (*p* > 0.05).

### 3.3. Effects of HILM on Serum Immunoglobulin, Complement c3, Complement c4 and Jejunum SIgA

As shown in Figure 3, compared with the control group, serum IgG and complement C3 were extremely significantly increased (*p* < 0.01) in the 1% HILM group, and serum Ig2a, IgG3 was also significantly up-regulated (*p* < 0.05).

### 3.4. Effects of HILM on Intestinal Morphology

As shown in Figure 4 and Figure 5, the addition of 1% HILM significantly increased the VH and VH/CD of each intestinal segment compared to the control group (*p* < 0.05), but there was no significant change in CD.

### 3.5. Effects of HILM on Spleen Immune-Related Genes and Jejunum Barrier-Related Gene mRNA Expression

As shown in Figure 6a,b, compared with the control group, the addition of 1% HILM significantly up-regulated the mRNA expression of jejunum Occludin and the spleen IL-6 gene (*p* < 0.05) and extremely significantly up-regulated the mRNA expression of the CD4 gene in the spleen (*p* < 0.01).

### 3.6. Effects of HILM on Microflora in Cecal Contents

As shown in Figure 7a, there were 597 operational taxonomic units in the two groups of cecal microbiota. The specific operational taxonomic units of the control group and 1% HILM groups were 118 and 207, respectively. The indicators of α-diversity, including Chao 1 and the Shannon index, showed no significant differences in the treatment (Figure 7b-e), while β-diversity analysis evaluated by PCoA and PCA plots (Figure 7f) showed that HILM supplementation changed the microbial composition, but there was no significant difference (*p* > 0.05).

The effects of HILM on microbial composition at different levels are shown in Figure 8a–d. Compared with the control group, a total of 18 phyla were identified in the 1% HILM group (Figure 8a), among which Bacteroidetes and fixed bacteria were the main phyla, and the amount of *Campilobacterota* was significantly down-regulated (*p* < 0.05). At the family level, 20 families including *Rikenellaceae*, *Lachnospiraceae* and *Barnesiellaceae* were the most abundant families (Figure 8b), the abundance of *Barnesiellaceae* was significantly down-regulated (*p* < 0.05), and the abundance of *Lactobacillaceae* was significantly up-regulated (*p* < 0.05). Figure 8c shows the top 20 microflora in the cecal contents of geese at the genus level, and the abundance of *Barnesiella* was significantly decreased in the 1% HILM group compared with the control group (*p* < 0.05). In addition, HILM supplementation significantly up-regulated the abundances of *Enterolyticus*, *Lactobacillus listeri*, and *Lactobacillus vaginalis* and significantly down-regulated the abundance of *Enterobacter jejuni* (Figure 8d). The LDA score showed that 9 different species were obtained in the control group, while 15 different species were obtained in the 1% HILM group (Figure 8d). The cladogram based on LEfSe analysis (Figure 8e) showed that *Oscillospirales*, *Lactobacillaceae* and *Butyricicoccaceae* were significantly different and played important roles in the HILM group.

## 4. Discussion

With the rapid growth of the global population, there is an increasing demand for animal protein resources, but the resources such as water and arable land for the production of feed ingredients are scarce. Therefore, there is an urgent need to develop new alternatives or supplements for feed ingredients. A large number of previous studies have confirmed that HILM can improve the production performance of fish, ducks, laying hens, broilers, pigs and other animals [18,19,20,21,22,23]. Similarly, we demonstrated in Experiment I that the addition of 1% HILM significantly increased ADG in Sichuan White Geese, suggesting that HILM can be used as a growth promoter for geese.

Poultry are natural hosts for AIV, which can cause the virus to persist in the host for a longer period of time, affecting waterfowl production and contaminating the environment [24,25]. AIV is difficult to prevent and control, it is easy to mutate and produce various subtypes, vaccination remains one of the most effective ways to prevent AIV. Therefore, antibody titers can demonstrate the antibody concentration in the body of Sichuan White Geese after vaccination, which is a reflection of their defense capacity [26]. At present, animals in intensive farming models are vulnerable to environmental factors, nutritional levels, immune stress and long-distance transportation leading to a decline in immunity [27]. HILM can broadly regulate the immune system [28] and reduce the impact of intensive farming on animals. Our study showed that the addition of 1% HILM significantly increased the serum strain titers of H7-R4 and H5-R14 at 33 d, and H5-R13 at 40 d. In addition, the addition of 1% HILM to the diet enhanced the AIV-specific antibody levels at all periods after vaccination. In conclusion, the addition of 1% HILM enhanced the immune response of the AIV vaccine in Sichuan White Geese, and this dose was also the optimal dosage under the conditions of experiment I.

IgG1 is the most prevalent subclass of immunoglobulin G, while IgG2a is primarily responsible for the immune response to polysaccharide antigens. These antibodies are produced in response to the activities of Th2 and Th1 cell reactions, respectively [29]. Additionally, serum complement C3 and C4 play crucial roles in eliminating pathogenic microorganisms through various effector functions [30]. In this study, HILM supplementation significantly up-regulated serum IgG, IgG1, IgG2a, IgG3 and complement C3 compared to the control group, indicating that HILM could maintain the balance of Th1/Th2 cell responses, which is conducive to the stabilization of the body’s immune system.

As the largest secondary lymphoid organ in the body, the spleen plays a key role in the immune system, and its secretion levels of cytokines are an indirect reflection of immune function. IL-6 is a key inflammatory cytokine, and CD4 molecules transmit signals that promote T-cell activation and enhance the immune response of T lymphocytes [31,32]. The results of this study showed that the relative mRNA expression levels of IL-6 and CD4 genes in the spleen of the 1% HIMM group were significantly increased compared with the control group. For birds, an adequate supply of energy and nutrients is essential for optimal immune function, and HILM has been demonstrated to serve as a digestible source of amino acid for birds, supporting the biosynthetic needs required for T lymphocyte proliferation and differentiation, as well as promoting the proliferation of CD8+ T lymphocytes [33]. These results inferred that dietary HILM supplementation improved the immune function of Sichuan White Geese by up-regulating the mRNA expression of IL-6 and CD4 genes in the spleen.

Intestinal health is closely related to intestinal morphology, with increased intestinal villus height enhancing nutrient absorption, while decreased crypt depth indicates higher cell maturity and improved secretory function. The ratio of villus height to crypt depth (VH/CD) serves as a comprehensive indicator of intestinal digestion and growth status [34,35,36,37]. Our study showed that the addition of 1% HILM significantly increased VH and VH/CD in the duodenum, jejunum and ileum, suggesting that HILM could maintain normal intestinal morphology, so as to enhance the nutrient absorption capacity of Sichuan White Geese. This finding aligns with previously reported results in black-boned chickens [38]. Furthermore, the improvement in the intestinal morphology of Sichuan White Geese observed in this study may be attributed to the presence of lauric acid in HILM; however, the specific mechanism of action requires further investigation [39].

SIgA primarily exerts its mucosal immune effects through numerous non-inflammatory pathways. In this study, the addition of 1% HILM significantly increased the SIgA levels in the jejunal mucosa. This finding is consistent with results reported by Chu et al. [23], which demonstrated that replacing soybean meal with full-fat black soldier fly larval powder significantly increased SIgA levels in the ileum of laying hens. The proposed mechanism involves chitin in the black soldier fly larvae, which, upon decomposition, can bind to pathogen sites, triggering the release of SIgA into the intestinal lumen. This process may inhibit the secretion of pro-inflammatory factors and enhance the immune function of the intestinal mucosa [40,41]. Moreover, Claudin, Occludin and ZO-1 genes regulate the expression of tight junction proteins between intestinal epithelial cells, which contributes to the maintenance of intestinal barrier function [42,43]. Dietary HILM supplementation could up-regulate intestinal barrier genes ZO-1, Occludin and Mucin-1 in weaned piglets and fattening pigs [44]. In the present study, we also observed a significant up-regulation of Occludin mRNA expression in the jejunum of Sichuan White Geese after HILM supplementation. In short, dietary HILM supplementation improved intestinal development and intestinal mucosal immune barrier function in Sichuan White Geese.

The gut microbial community is complex, diverse, and maintained in a dynamic equilibrium, playing a vital role in promoting nutrient digestion and absorption of nutrients and stimulating the host immune response. The stabilization of intestinal flora is beneficial for the health and growth performance of organisms. However, toxic metabolites produced by intestinal microorganisms may adversely affect the host directly or indirectly, potentially triggering intestinal inflammation [45,46,47,48]. Dietary HILM substitution or supplementation can positively affect the cecal microflora of broilers via increasing the abundance of beneficial microorganisms [3]. In this study, dietary HILM supplementation changed the composition of the cecal microbiota and significantly increased OTUs of microflora in Sichuan White Geese. Further investigation showed that HILM significantly decreased the relative abundance of *Campilobacterota*, *C.jejuni* and *Barnesiaceae* and significantly increased the relative abundance of Lactobacillus and *L.amylovorus*. LEfSe analysis showed that *Oscillospirales*, *Lactobacillaceae* and *Butyricicoccaceae* played important roles in the cecal microflora. *Campylobacter* is an opportunistic pathogen that can cause gastroenteritis. Previous studies have shown that intestinal flora such as *Oscillospira*, *Butyricococcus* and *Lactobacilli* can produce short-chain fatty acids (SCFAs), which contribute to the development of the intestinal barrier and reduce intestinal inflammation [49]. However, *Campylobacter jejuni* consumes SCFA as a carbon source for its colonization and disrupts the intestinal mucosal barrier, causing the birds’ gut to exhibit high permeability [50]. Studies also have found that *Lactobacillus amylovorus* enhances the synthesis of host endogenous defense peptides, which is beneficial to body health and growth performance [51]. *Lactobacillus* exhibits strong antibacterial activity and inhibits the growth of pathogens in the gastrointestinal tract through competitive exclusion, thereby reducing the risk of opportunistic bacterial infections in geese [52]. Some scholars have pointed out that an increase in Lactobacillus was also observed when broilers and finishing pigs were fed dietary supplementation of HILM [18,53]. These results suggested that dietary HILM supplementation improves the abundance and composition of cecal microflora, which in turn is beneficial for the growth performance and intestinal mucosal barrier of Sichuan White Geese.

## 5. Conclusions

In conclusion, dietary supplementation with HILM improved the growth performance and enhanced the immune response of AIV vaccination in Sichuan White Geese. The underlying mechanisms may include the up-regulation of serum immunoglobulin levels, improvement of intestinal morphology, maintenance of the intestinal barrier, and regulation of the abundance and composition of cecal microflora. These findings highlight the potential application of HILM in geese production.

## Figures and Tables

**Figure 1 vetsci-11-00615-f001:**
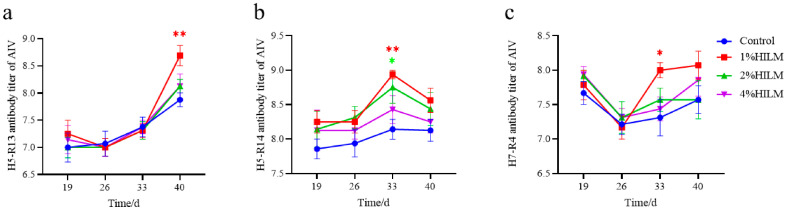
Effect of HILM on H5-R13 titer (**a**), H5-R14 titer (**b**) and H7-R4 titer (**c**) of AIV specific antibody in Sichuan White Geese. * indicates that there is a significant difference compared with the control group (*p* < 0.05), and ** indicates an extremely significant difference compared with the control group (*p* < 0.01).

**Figure 2 vetsci-11-00615-f002:**
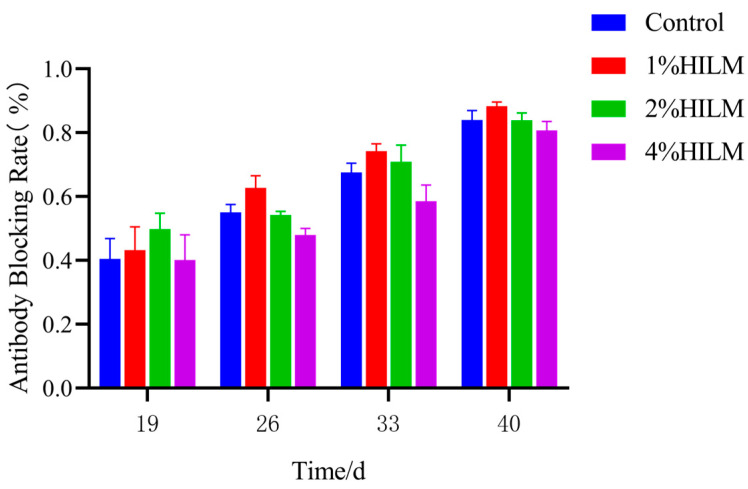
Effect of HILM on AIV-specific antibody blocking rate of Sichuan White Geese.

**Figure 3 vetsci-11-00615-f003:**
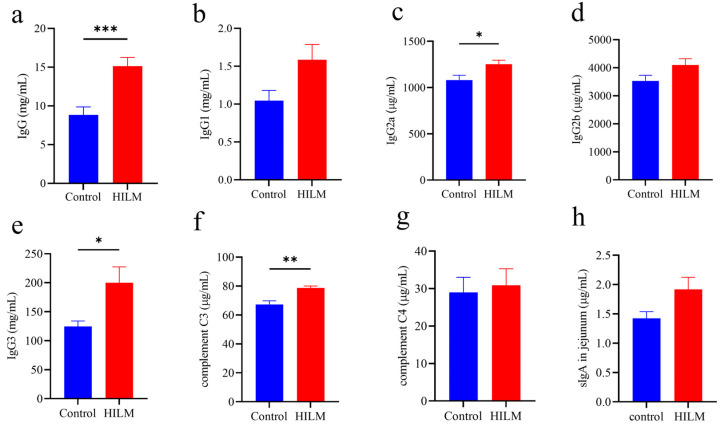
Effect of HILM on serum IgG (**a**), IgG1 (**b**), IgG2a (**c**), IgG2b (**d**), IgG3 (**e**), complement C3 (**f**), complement C4 (**g**), and SIgA in jejunum (**h**) of Sichuan White Geese. * indicates that there is a significant difference compared with the control group (*p* < 0.05), *** and ** indicates an extremely significant difference compared with the control group (*p* < 0.01).

**Figure 4 vetsci-11-00615-f004:**
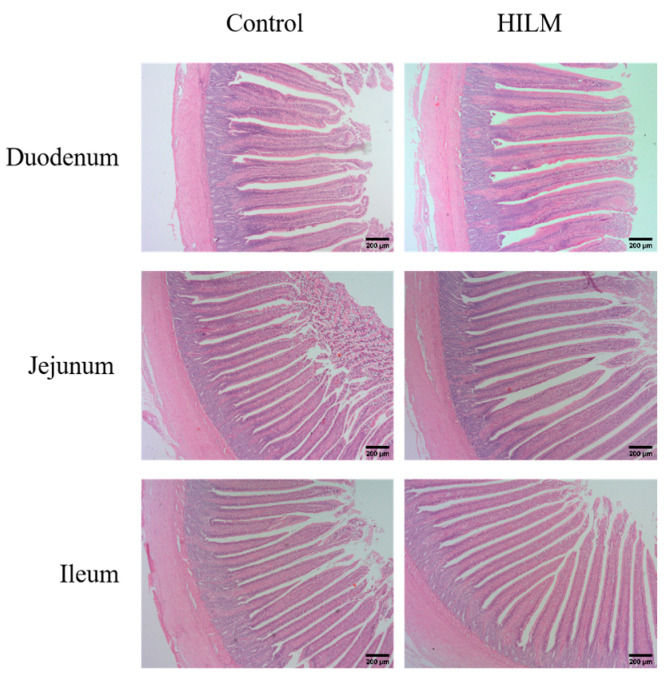
Effect of HILM on intestinal morphology of Sichuan White Geese by HE staining.

**Figure 5 vetsci-11-00615-f005:**
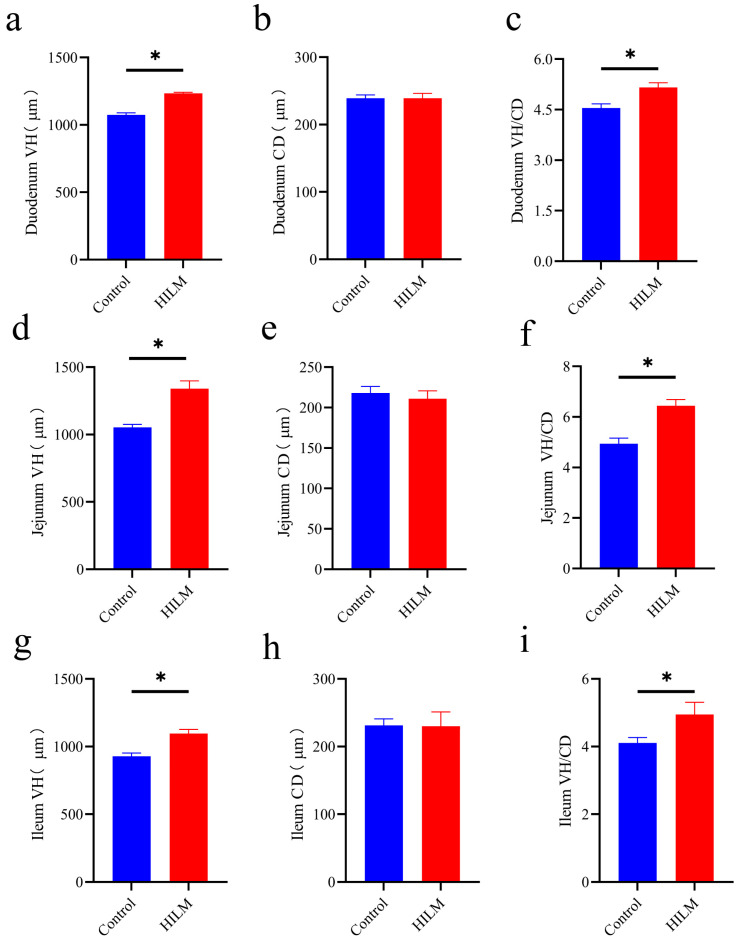
Effect of HILM on intestinal morphology (VH, CD and VH/CD) of duodenum (**a**–**c**), jejunum (**d**–**f**) and ileum (**g**–**i**) in Sichuan White Geese. * indicates that there is a significant difference compared with the control group (*p* < 0.05).

**Figure 6 vetsci-11-00615-f006:**
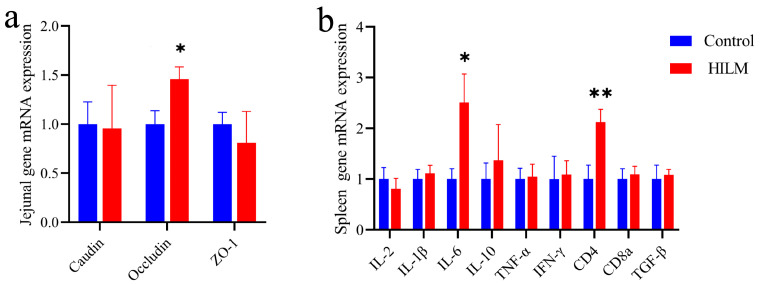
Effect of HILM on jejunum (**a**) and spleen gene mRNA expression (**b**) of Sichuan White Geese. * indicates that there is a significant difference compared with the control group (*p* < 0.05), and ** indicates an extremely significant difference compared with the control group (*p* < 0.01).

**Figure 7 vetsci-11-00615-f007:**
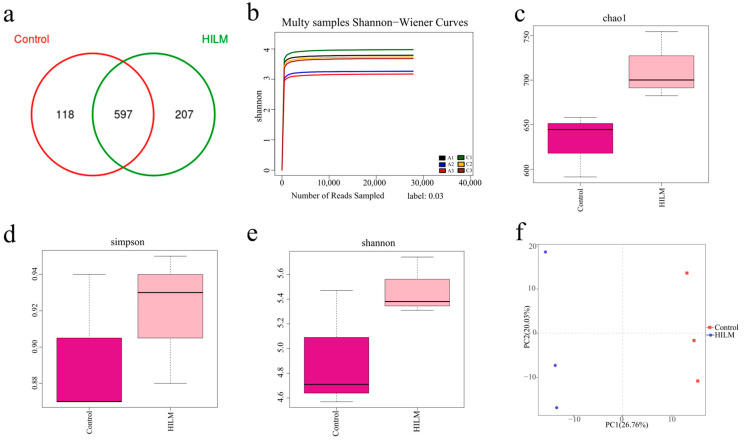
Diversity analysis of HILM impacts on Sichuan White Geese. (**a**) Venn diagram; (**b**–**e**) alpha diversity index as accessed by Chao1, Simpson and Shannon indices; (**f**) beta diversity as accessed by principal coordinate analysis (PCoA).

**Figure 8 vetsci-11-00615-f008:**
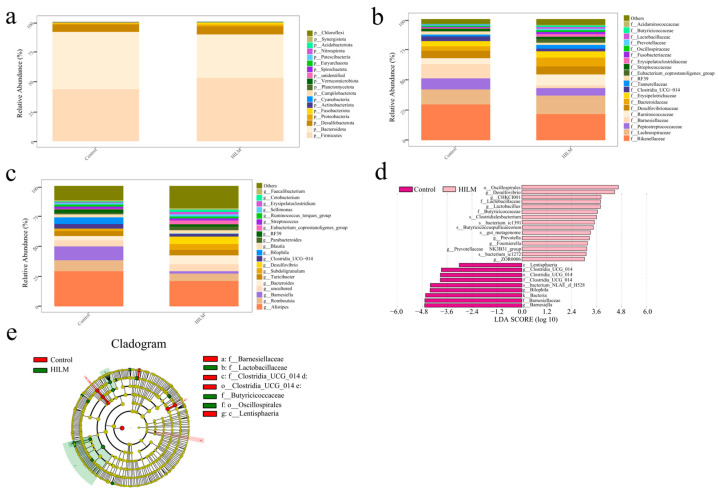
Effects of HILM on microflora in cecal contents. (**a**–**c**) The relative abundance of top 20 bacteria at the phylum, family and genus levels; (**d**) LDA score of LEfSe analysis; (**e**) evolutionary branching diagram of LEfSe analysis (Control group was marked in Red and HILM group in Green).

**Table 1 vetsci-11-00615-t001:** Measured nutrient composition of the HILM.

Items	Content (%)
Crude protein	42.69
Ether extract	27.80
Crude fiber	5.00
Calcium	4.72
Total phosphorus	0.83
Isoleucine	1.63
Leucine	2.56
Histidine	1.33
Lysine	2.40

**Table 2 vetsci-11-00615-t002:** Experimental diet composition and nutritional level (air-dried basis).

Items	Content (%, Unless Otherwise Indicated)			
	Control	1% HILM	2% HILM	4% HILM
Ingredients				
Corn	65.40	65.40	65.40	65.40
Soybean meal	28.00	28.00	28.00	28.00
Alfalfa meal	3.00	3.00	3.00	3.00
HILM	0.00	1.00	2.00	4.00
Limestone	1.20	1.20	1.20	1.20
Ca(HCO_3_)_2_	0.80	0.80	0.80	0.80
NaCl	0.33	0.33	0.33	0.33
Lysine	0.19	0.19	0.19	0.19
Methionine	0.08	0.08	0.08	0.08
Premix ^1^	1.00	1.00	1.00	1.00
Nutrient levels ^2^				
Metabolizable energy (kcal/kg)	2811.00	2797.00	2800.00	2800.00
Crude protein	18.12	18.03	18.01	18.00
Calcium	0.80	0.81	0.80	0.80
Total phosphorus	0.51	0.51	0.51	0.51
Available phosphorus	0.31	0.31	0.30	0.30

^1^ The premix provided the following per kg of diet: vitamin A 10,000 IU, vitamin D 3000 IU, vitamin E 25 IU, vitamin K 2.5 mg, vitamin B1 3 mg, vitamin B2 12 mg, vitamin B6 8 mg, vitamin B12 0.08 mg, pantothenic acid 25 mg, nicotinic acid 80 mg, folic acid 3 mg, biotin 0.25 mg, choline chloride 1.5 g, Cu (CuSO_4_·5H_2_O) 0.015 g, Fe (FeSO_4_·H_2_O) 85 mg, Zn (ZnSO_4_·H_2_O) 80 mg, Mn (MnSO_4_·H_2_O) 85 mg, I 0.5 mg, Se 0.3 mg. ^2^ Crude protein, calcium, and total phosphorus were measured values, and the others were calculated values.

**Table 3 vetsci-11-00615-t003:** Primer sequence information.

Gene	Primer Sequence (5′→3′)	Product Size (bp)
*ZO-1*	F: GACCATTCCAGACATTCTCCACAGC R: TCGCCTGCCACCTCTTCCATAG	146
*Occludin*	F: ACAGCAGCAGCACTTACCTCAAC R: AGGCAGAGCAGGAGGACGATG	109
*Claudin-1*	F: GACCAGGTGAAGAAGATGCGGATG R: CGAGCCACTCTGTTGCCATACC	107
*IL-1β*	F: GCACAAGGACTTCGCCGACAG R: GAAGGACTGGGAGCGGGTGTAG	130
*IL-2*	F: AACGGGATGCAAGATCTGTGAAGC R: ATGTGAGAAAGTTGGTCAGCTCTCG	80
*IL-6*	F: AAGCATCTGGCAACGACGATAAGG R: TGTGAGGAGGGATTTCTGGGTAGC	90
*IL-10*	F: TGCCAGTCGGTGTCGGAGATG R: CTGGTGGTGCTCGCTGTTCTTG	81
*TNF-α*	F: CTGGCTAAGACCGTGGTCAGTTTC R: GGTGACGCTGAATGATCTGGTGAAG	115
*IFN-γ*	F: GAAGTTCAAAGACCTCGTGGACCTG R: AACAGCTCACTCACAGCCTTGC	81
*CD4*	F: GCTGGTGTGTTGATGTTTGTCCTTG R: GCTGTCTTGCTCGTGCCATCC	103
*CD8a*	F: ACGAGGCAGAGACGAGCAAGG R: CCAGGGCAATGAGAAGCAGGATG	102
*TGF-β*	F: TTCCAACACCAGGTCCTACTCCAG R: GCAGACAGGTCCGGCAATAACAG	86
*β-actin*	F: GCACCCAGCACGATGAAAAT R: GACAATGGAGGGTCCGGATT	150

**Table 4 vetsci-11-00615-t004:** Effect of HILM on growth performance of Sichuan White Geese.

Items	Group			
	Control	1% HILM	2% HILM	4% HILM
ADG, g/d	42.65 ± 3.63 ^b^	48.13 ± 3.75 ^a^	45.95 ± 4.81 ^b^	44.71 ± 3.59 ^b^
ADFI, g/d	95.94 ± 7.79	104.66 ± 8.94	103.55 ± 8.37	103.50 ± 7.74
FCR, (g:g)	2.30 ± 0.05	2.15 ± 0.05	2.23 ± 0.07	2.30 ± 0.06

Note: Means with different lowercase superscripts in the same column differ significantly (*p* < 0.05), and those with the same lowercase or no lowercase superscripts in the same column differ not significantly (*p* > 0.05).

## Data Availability

All the data presented in the study are included in the article; further inquiries can be directed to the corresponding authors.

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
