# Peer review of "Hermetia illucens Larvae Meal Enhances Immune Response by Improving Serum Immunoglobulin, Intestinal Barrier and Gut Microbiota of Sichuan White Geese After Avian Influenza Vaccination"

_vetsci, 2024, doi:10.3390/vetsci11120615_

Round 1
Reviewer 1 Report
Comments and Suggestions for Authors
The work is not innovative but is reasonably well-structured and described. The article doesn't require major revisions; the only problem, in my opinion, is the limited novelty of the topic.
Author Response
Dear Reviewer:
Thank you very much for being able to review my manuscript in your busy schedule.
I have written my answer in the word. Please see the attachment.
Look forward to hearing from you and wish you a nice day!

Reviewer 2 Report
Comments and Suggestions for Authors
The manuscript submitted by Xie et al., described the immune enhancement function of Hermetia Illucens Larvae Meal on Geese after Avian Influenza Vaccination. The data showed that 1% HILM could stimulate the antibody level, and increased the antibody subtypes production, and also regulated the distribution of microbiota in geese. The results of manuscript has certain clinical application reference value.
However, there are some shortcoming in the manuscript.
1. in Experiment I and Experiment II, the number of goslings is different. What is the basis for grouping differences in quantity?
2. When conducting differential analysis, the control group was not explained. Which group are each experimental group compared and analyzed with?
3. In Figure 2, There is no significant difference in antibody levels during a specific time period, and the antibody levels gradually increase at different time periods. Can it be understood that as the immune time prolongs, antigen induced antibodies in the body gradually increase, which have little relation with HILM?
4. The clarity of Figure 4 is too low, and the image is not clear.
5. Figure 5~8, Missing image titles.
Author Response

(The authors gave the same response as above.)

Reviewer 3 Report
Comments and Suggestions for Authors
Manuscript vetsci-3274715 investigated the effects of dietary addition of different doses of Hermetia Illucens Larval meal on the immune response of geese after AIV vaccination, and made a preliminary investigation of the mechanism in terms of serum immunoglobulin, spleen immunity-related genes, intestinal barrier and intestinal microbiota. The experimental design is more reasonable, and the application of Hermetia Illucens on geese is less reported in the research, which is innovative and meaningful. There are still some deficiencies in the writing of the article, and we hope to give modifications.
1.Line 67, “ETEC K88” does not require an abbreviated format.
2.“Hermetia illucens” may be replaced by the abbreviation in all but the first occurrence, e.g. line 56 and line 80.
3.Check the formatting throughout the text. For example, there should be a space in the middle of -80℃ in line 132, make sure that the units are uniform, for example, “day and d”. Please check the entire manuscript thoroughly.
4.Line 115, there is no need to add “s” to indicate the time point.
5.In line 259, “SIgA in jejunum (e)” should be placed before “IgG3 (f)” in serial number.
6.In general, levels above “genus”, e.g., “order” and “family”, are written in block letters, and genus and species names are italicized, with the first letter of the genus capitalized and the first letter of the species name not capitalized. Please read through the text and make the corrections.
7.Line 365 mentions “jejunal SIgA secretion”, but I didn't find it in the article, please add.
8.What was the basis for the selection of the AIV vaccine subtypes mentioned in the manuscript?
Author Response

(The authors gave the same response as above.)

Round 2
Reviewer 2 Report
Comments and Suggestions for Authors
The author has made detailed revisions to the proposed reviewer‘s comments. The design of revised manuscript is reasonable and the data has certain reference value. I agree to accept the resubmitted manuscript.
Author Response
Dear Reviewer,
Thank you for your support! Thank you very much for your suggestions and comments on our manuscripts! Have a great day! Thank you!
Best Regards!